# A Narrative Review on Non-Cirrohotic Portal Hypertension: Not All Portal Hypertensions Mean Cirrhosis

**DOI:** 10.3390/diagnostics13203263

**Published:** 2023-10-20

**Authors:** Michele Fiordaliso, Giuseppe Marincola, Barbara Pala, Raffaella Muraro, Mariangela Mazzone, Maria Carmela Di Marcantonio, Gabriella Mincione

**Affiliations:** 1Department of Medicine and Ageing Sciences, University “G. D’Annunzio” of Chieti–Pescara, Via dei Vestini 29, 66100 Chieti, Italy; michele.fiordaliso@gmail.com; 2Bariatric and Metabolic Surgery Unit, Fondazione Policlinico Universitario Agostino Gemelli IRCCS, Largo Agostino Gemelli 8, 00168 Rome, Italy; giuseppe.marincola@policlinicogemelli.it; 3Division of Cardiology, Department of Clinical and Molecular Medicine, Sant’Andrea Hospital, Sapienza University of Rome, Via di Grottarossa, 1035/1039, 00189 Rome, Italy; barbara.pala93@gmail.com; 4Department of Innovative Technologies in Medicine & Dentistry, University “G. D’Annunzio” of Chieti–Pescara, Via dei Vestini 29, 66100 Chieti, Italy; raffaella.muraro@unich.it (R.M.); mariangela.mazzone@unich.it (M.M.); dimarcantonio@unich.it (M.C.D.M.)

**Keywords:** non-cirrhotic portal hypertension (NCPH), idiopathic non-cirrhotic portal hypertension (INCPH), idiopathic portal hypertension, non-cirrhotic portal fibrosis, hepatoportal sclerosis

## Abstract

Non-cirrhotic portal hypertension (NCPH), also known as idiopathic non-cirrhotic portal hypertension (INCPH) and porto-sinusoidal vascular disorder (PSVD), is a rare disease characterized by intrahepatic portal hypertension (IPH) in the absence of cirrhosis. The precise etiopathogenesis of IPH is an area of ongoing research. NCPH diagnosis is challenging, as there are no specific tests available to confirm the disease, and a high-quality liver biopsy, detailed clinical information, and an expert pathologist are necessary for diagnosis. Currently, the treatment of NCPH relies on the prevention of complications related to portal hypertension, following current guidelines of cirrhotic portal hypertension. No treatment has been studied that aimed to modify the natural history of the disease; however, transjugular intrahepatic porto-systemic shunt (TIPS) placement, shunt and liver transplantation are considerable symptomatic options. In this review, we discuss the heterogeneity of NCPH as well as its etiopathogenesis, clinical presentation and management issues. Starting from the assumption that portal hypertension does not always mean cirrhosis, cooperative studies are probably needed to clarify the issues of etiology and the possible genetic background of this rare disease. This knowledge might lead to better treatment and perhaps better prevention.

## 1. Introduction

Non-cirrhotic portal hypertension (NCPH) is a rare disease characterized by portal hypertension, splenomegaly, hypersplenism, and pancytopenia. It is not associated with cirrhosis on liver histology or other known liver diseases [1]. The condition was initially described in 1889 by an Italian pathologist named Banti [2], and it has been referred to by different names over time such as non-cirrhotic portal fibrosis [3], hepatoportal sclerosis [4], and non-cirrhotic portal hypertension [5,6]. NCPH is diagnosed after excluding other causes of portal vein or hepatic venous outflow tract obstruction. The disease progresses through different phases with symptoms ranging from splenomegaly and anemia to ascites and life-threatening complications such as gastrointestinal bleeding [2].

The incidence of NCPH varies globally, but it is considered rare. The exact cause of NCPH is unknown, and there are no specific laboratory tests to confirm its presence. Radiological imaging shows characteristic liver changes, and esophageal varices and bleeding are common complications. Older studies suggest that NCPH accounts for a small percentage of cases of portal hypertension, and its incidence may be even lower than previously thought.

This summary also highlights the lack of consensus and knowledge gaps regarding NCPH, including its underlying causes and clinical presentation. The review aims to provide an overview of the current understanding of NCPH and discusses the standard of care and recent advancements in its management. It evaluates different therapies that have been studied to reduce the risk of complications associated with NCPH.

Our narrative synthesis emphasizes the importance of recognizing that portal hypertension does not always indicate cirrhosis in clinical practice. By increasing knowledge about NCPH and applying evidence-based approaches to prevent and treat its complications, there is a potential for improving clinical outcomes and patient well-being.

## 2. Epidemiology

NCPH has varying prevalence worldwide with a higher occurrence in developing countries, particularly in Asia, notably Japan and India. In relation to the incidence in these countries, we observed that the highest incidence was reported in Japan in 1975 and was 31% in patients admitted to Nagoya University Hospital with NCPH [7]. India also had a high incidence of NCPH in 1980 (range 7.9 to 46.7%) with a slight decrease in subsequent years (range 6.6 to 41%) [8].

The reasons for these geographic differences are not fully understood, but at least with regard to the high incidence in India, it seems to be associated with poor socioeconomic conditions [8]. Most reports on non-cirrhotic portal fibrosis in India have shown a male predominance, with a male:female ratio of 2:1 to 4:1 and a mean age of 30–35 years.

In contrast, in Japan, there is a female predisposition [9]. In Western countries, the prevalence of NCPH is much lower, ranging from 3 to 6% of patients with portal hypertension, and this is slightly more common in males than females [10,11,12,13,14]. However, gender differences are difficult to explain. There is speculation about a decreased incidence of the disease, which is probably related to improved standards of perinatal hygiene and care that would lead to reduced incidence of umbilical sepsis and diarrheal episodes in early childhood [8].

## 3. Pathophysiological Differences between NCPH and Cirrhosis

Although both diseases involve an increase in portal venous pressure, in cirrhosis, the increase in resistance is at the sinusoidal level [15] with a rise in arterioportal (AP) shunts and arterial blood flow, whereas in NCPH, resistance is mainly increased at the presinusoidal level (Figure 1) with a reduction in arterial flow, while the number of AP shunts is negligible (Table 1). The elevation of portal venous pressure in NCPH is attributed to histological changes in the portal vein branches, including sclerosis, narrowing or dilation of the lumen, occlusion of the lumen and herniation in the lobule [16]. This lesion has been reported previously under different names, including hepatoportal sclerosis, phlebosclerosis or portal vein obliteration. The term “portal vein stenosis” is proposed to replace obliterative portal venopathy (specific histological sign) as it encompasses a range of histological features: from reduced luminal size to complete obliteration or disappearance of portal vein branches. In a normal liver, the angles between the hepatic vein and its tributaries are wide; some are almost at right angles. The tributaries gradually taper as they approach the periphery. In idiopathic portal hypertension, the angles between the hepatic vein and its tributaries are narrow, and these appear very close together with an irregular and tortuous appearance (non-specific histological signs). In cirrhosis, on the other hand, the angles appear obtuse and, unlike NCPH, anastomoses between the hepatic veins are very rarely found [17]. These anastomoses tend to increase as we approach the periphery, which is probably related to the loss of parenchyma due to portal circulatory insufficiency. In cirrhosis, moreover, retrograde flow in the portal venous system is common.

Another detectable feature in patients with NCPH is the presence of incomplete fibrous septae (specific histological sign). These are thin fibrous septae that cross the hepatic parenchyma, creating a nodular architecture with an approximation of the hepatic vein to the portal tract. It is a complex entity that is not only difficult to diagnose but which is also part of the characteristics in common with the picture of regressing cirrhosis [18].

Another aspect to consider in the biopsy is the possibility that the liver develops regenerative nodules of hyperplasia (specific histological sign) due to an alteration of the balance between portal and arterial flow. Hepatocellular adenomas and carcinomas have been described in this context [19].

A staging system has been developed by Nakanuma et al. to assess the severity of IPH [20]. It comprises four stages that reflect the extent of peripheral parenchymal atrophy and the presence of obstructive thrombosis [9].

Considering only the histological characteristics, this pathology has assumed numerous names in different countries such as “non-cirrhotic portal fibrosis” (in India), “idiopathic portal hypertension” (in Japan), and “hepatoportal sclerosis” (in Western countries) along with many other names such as “non-cirrhotic intrahepatic portal hypertension”, “nodular regenerative hyperplasia”, and “obliterative portal venopathy”. Given the involvement mainly at the portal level and the presinusoidal area, the Vascular Liver Diseases Interest Group (VALDIG) coined the term porto-sinusoidal vascular disorder (PSVD) to describe patients with non-cirrhotic portal hypertension and with these histological changes detected on biopsy [21]. The term PSVD replaced the term INCPH in order to include the patients with specific histological features but no clinical signs of portal hypertension.

## 4. Pathophysiology

Two speculative theories exist to explain the pathophysiology of NCPH. The first theory suggests that initial injury to the intrahepatic vascular bed triggers increased resistance to portal blood flow. This is believed to be caused by portal venopathy resulting from factors like hypercoagulability [22,23], endothelial damage, or autoimmune damage. Experimental studies in animals and histologic findings in humans support the vascular cause, where the occlusion of portal veins leads to liver ischemia [24], the atrophy of vulnerable regions, and the compensatory hypertrophy of periportal regions [25,26]. Autoimmune mechanisms and T-cell-mediated endothelial cell injury have also been implicated [27]. Antiphospholipid antibodies are found in a significant number of NCPH patients, indicating a potential autoimmune antibody-dependent mechanism [22].

The second theory regarding NCPH suggests that an initial event triggers the dilation of the splenic sinuses, leading to splenomegaly and subsequent increased portal venous flow, resulting in elevated portal pressure. Numerous studies have indicated that the underlying cause of NCPH is not associated with hepatic abnormalities but rather with increased portal venous flow due to splenomegaly [28,29].

An overproduction of nitric oxide has been designated responsible for both splenic sinuses dilatation and massive splenomegaly in patients with NCPH [30]. Liver samples in these patients show normal histopathology. Splenectomy-induced disease remission supports the role of splenomegaly in NCPH development. Advanced cases may involve intrahepatic resistance due to portal venous microcirculation obliteration, which is potentially caused by thrombophilia, immunologic disorders, or infection [31,32,33]. A challenge to this theory is that normal livers are able to accommodate marked increases in portal blood flow with a minor or no increase in portal pressure; for this reason, this theory currently remains poorly accredited.

## 5. Etiology and Risk Factors

The etiology and risk factors of NCPH can be classified into five categories: immunological disorders, prothrombotic conditions, chronic infections, exposure to medications or toxins and genetic predisposition. NCPH seems to be a multifactorial disease in which two or more etiological factors may play a role, and further research is needed to fully understand its pathogenesis.

### 5.1. Immune-Related Factors

Immunological factors play a role in NCPH, with autoimmune diseases like mixed connective tissue disease, systemic sclerosis, and systemic lupus erythematosus being associated with NCPH. Anticardiolipin antibodies and an increased expression of certain immune-related molecules have been observed in NCPH patients [1,31,32,34,35,36].

The detection of anticardiolipin antibodies, as indicated by positive lupus results, implies the existence of a shared immunological mechanism underlying the development of both NCPH and SLE. Furthermore, the presence of Raynaud’s phenomenon, coupled with the presence of anti-ribonuclear protein antibodies, appears to be correlated with NCPH. In NCPH, there is a higher occurrence of human leukocyte antigen-DR (HLA-DR) antigen expression on the microvasculature of portal tracts. This antigen plays a role in immune recognition and various immunological reactions, potentially triggering the immune attack on portal microvessels in NCPH. The elevated presence of interferon in portal venous blood might be responsible for this increased HLA-DR expression [37].

Furthermore, some NCPH patients exhibit increased levels of soluble vascular cell adhesion molecule 1 (VCAM-1) in serum [37], which could indicate an immunopathological event contributing to the development of NCPH.

### 5.2. Coagulation Disorders

Coagulation disorders may contribute to the development of IPH, as a hypercoagulable state has been implicated. Portal vein thrombosis has been observed in NCPH patients, particularly those with overt prothrombotic conditions [5,38].

It is not clear whether portal vein thrombosis is a complication of NCPH, whether it contributes to the pathogenesis of the disease or whether both theories are valid. In fact, the presence of a thrombophilic factor and the NCPH-induced decrease in portal flow may contribute to PVT. On the other hand, the imbalance between portal and arterial flow induced by PVT can cause the sinusoidal vascular changes found in NCPH. Further studies are needed [39].

### 5.3. Infectious Etiology

Data indicate that intestinal infection with *Escherichia coli* (E. coli) might cause recurrent septic embolization leading to endothelial damage and the obstruction of small portal veins, which is a probable trigger of NCPH [40,41]. The high prevalence of NCPH in low socioeconomic areas with a high rate of abdominal infections in early childhood lends credit to this theory. Frequent occurrences of umbilical sepsis, bacterial infections, and diarrhea during childhood, especially in socioeconomically deprived populations, increase the risk of developing NCPH [36,42].

Moreover, cases of HIV-infected patients who developed variceal bleeding, due to underlying non-cirrhotic portal hypertension, have been reported. Endothelial cell damage in the portal system causing veno-occlusion and underlying thrombophilic states are proposed mechanisms due to antiretroviral therapy (particularly long-term didanosine) and/or HIV [28,43].

### 5.4. Toxic or Chemical Agents

Azathioprine (AZA), a thiopurine analog, is an immunosuppressive agent that acts as an antagonist of purine metabolism via 6-thioguanine to disrupt the making of RNA and DNA by cells. AZA has recently been reported to cause NCPH, but the pathogenesis of AZA-induced NCPH remains uncertain [44].

Oxaliplatin is a chemotherapeutic agent widely used in chemotherapeutic regimens for colorectal carcinoma with direct toxic action on DNA, which leads to an arrest in its synthesis and induces cell death. Few studies showed how oxaliplatin use can trigger NCPH, producing direct damage to the liver or through fibrosis induction. However, in both cases, sinusoidal and post-sinusoidal damage occur [45].

Chronic exposure to toxic or chemical agents such as arsenic or vinyl chemicals has been associated with the development of NCPH. Exposure to these chemical substances for a long time may result in histological findings resembling hepatoportal sclerosis [1].

### 5.5. Genetic Predisposition

Genetic predisposition may also play a role with a mutation in the DGUOK gene identified in some cases of early-onset non-cirrhotic portal hypertension [1,46].

Mutations in the telomerase gene complex (mutations in TERT and TERC) seem to play a role in patients with NCPH [47].

## 6. Natural History of NCPH

In patients with IPH, the liver will undergo a gradual process of atrophy due to reduced blood supply to the periphery. Normally, liver function reserves are well preserved. The main complications faced by these patients are variceal bleeding, portal thrombosis, ascites and hepatic encephalopathy, which are the same as in patients with portal hypertension due to cirrhosis, and this is the reason why the management of IPH currently follows the same guidelines as cirrhosis. Although mortality from ruptured varices is low in patients with IPH, due to good liver function compared to cirrhotic patients, the leading cause of death is variceal bleeding [48].

In the patients with IPH, variceal progression is more rapid and bleeding is more frequent than in cirrhotics. In patients with IPH, the rate of development of varices at risk of bleeding was significantly higher than in patients with cirrhosis independently on the size of varices at the first endoscopy. Current guidelines state that patients with cirrhosis who do not have varicose veins at the first endoscopic check-up should repeat the examination after two or three years. Given the natural history of patients with IPH, we would like to suggest that follow-up should probably be brought forward in these patients.

Another relevant difference between cirrhotic and non-cirrhotic patients is represented by the incidence of portal vein thrombosis, which is expectedly higher in the patients with IPH, which have a spleen and portal vein diameter significantly higher than those of patients with cirrhosis. The development of portal vein thrombosis in patients with IPH may be a significant factor for poor prognosis [49].

This tends to develop especially in patients with a prothrombotic state [5], hence the need to investigate these patients from the point of view of coagulation disorders so as to be able to intervene early with drugs such as anticoagulants. The practical benefits of the management of portal vein thrombosis (PVT) to improve the clinical course of IPH should be elucidated in future studies. Unlike the aforementioned complications, the incidence of ascites is lower in patients with IPH and tends to be associated with PVT [50]. The appearance of ascites is normally a sign of clinical deterioration of patients with IPH.

Hepatic encephalopathy (HE) is a complication much less frequent than in cirrhosis, but it has been reported to occur in 32% of patients with IPH [51]. The development of HE in these patients is strictly related to the presence of large porto-systemic shunt that is either spontaneous or iatrogenic. In fact, the study published from Bissonnette et al. report that 31% of patients with IPH who underwent TIPS developed this condition following the stent’s placement. In most cases, it was transient and easily managed with medical treatment [52]. Contrarily to cirrhosis, the risk of developing a hepatocellular carcinoma is very low.

In conclusion, as can be seen from the differences between the two pathologies, in the patients with IPH, the use of the same therapies used for cirrhotic patients with portal hypertension may not be correct. However, we are aware that given the limited number of cases, it may be difficult to conduct randomized controlled trials.

## 7. Diagnosis

IPH diagnosis is challenging, as there are no specific tests available to confirm the condition. Typically, the diagnosis is based on the exclusion of other liver diseases. A liver biopsy is essential to establish a firm diagnosis and rule out other causes of portal hypertension.

Liver biopsy can also be obtained through the transjugular catheterization of the hepatic veins with the double advantage of being able to measure the hepatic venous pressure gradient (HVPG). In patients with NCPH and clinical signs of portal hypertension, HVPG may be normal or slightly elevated. This is attributable both to the fact that portal hypertension is pre-sinusoidal and to the presence of intrahepatic vein-to-vein communications found in over 50% of patients, which prevent an occlusion and ensure that the pressure is normal [53].

### 7.1. Histological Features

The definitive diagnosis of NCPH mandates a liver biopsy examination that is preferably longer than 20 mm and contains a minimum of 10 portal tracts. There are three primary histological lesions specific to NCPH:Obliterative Portal Venopathy: Previously referred to by several names like hepatoportal sclerosis, this lesion signifies the narrowing of the portal vein branch lumen to the extent of its complete disappearance and its substitution by fibrosis. As a result, portal tracts can appear fibrotic and hard to discern due to the veiled vein lumen. Various immunohistochemical methods can help identify such cases with portal vein stenosis being a significant independent predictor of NCPH.Nodular Regenerative Hyperplasia (NRH): This represents a diffuse micronodularity of the liver parenchyma devoid of fibrosis. A reticulin stain usually highlights NRH, revealing small hyperplastic hepatocyte nodules interspersed with atrophic plates, sometimes demonstrating signs of ischemic biliary metaplasia.Incomplete Septal Fibrosis/Cirrhosis: This complex entity shows liver parenchyma crisscrossed by thin, incomplete fibrotic bands. Despite being challenging to discern, this feature often correlates clinically with NCPH.

In scenarios lacking specific histological lesions and overt signs of portal hypertension, NCPH diagnosis necessitates non-specific signs of both portal hypertension and NCPH histology. These can range from herniated portal vein branches to sinusoidal dilation and mild peri-sinusoidal fibrosis. A crucial understanding is that these histological changes can also emerge in the absence of portal hypertension. Therefore, they might indicate a preclinical stage when observed with unidentified mild liver enzyme anomalies and no portal hypertension [21].

### 7.2. Clinical Implication

Clinical manifestations of NCPH include variceal bleeding, which is typically well tolerated due to preserved liver function. Late-stage NCPH may occasionally lead to ascites, which are usually triggered by factors like variceal bleeding or infections. Hepatopulmonary syndrome and portal vein thrombosis are also observed in some cases [54]. NCPH is characterized by splenomegaly, and the size of the spleen is often larger compared to cirrhosis (Figure 2). Dilated superficial abdominal veins may be present, and mild hepatomegaly is seen in some patients [55]. Laboratory tests reveal preserved liver function, but anemia, leukopenia, and thrombocytopenia are common due to hypersplenism.

Radiological imaging is not distinctive for NCPH and can be difficult to differentiate from cirrhosis. Liver surface smoothness or nodularity may resemble cirrhosis or nodular regenerative hyperplasia, but the combination of caudate lobe hypertrophy and segment IV atrophy, typically seen in cirrhosis, is rare in NCPH.

Transient elastography can help differentiate portal vein thrombosis due to cirrhosis from that in the context of NCPH. A liver stiffness value lower than expected for cirrhosis (i.e., <13 kPa) is often observed in NCPH [56,57].

A recent study showed that the ratio between spleen stiffness measurement (SSM) and liver stiffness measurement (LSM) was 0.940 (*p* < 0.001) for differentiating NCPH from cirrhosis. The study proposed 2 as a cut-off, with good sensitivity (86.5%), specificity (92.7%), and accuracy (89.7%) for the diagnosis of NCPH [58].

Another study showed that SSM of >35.4 kPa has 93% sensitivity, 60% specificity, and 91% negative predictive value in the diagnosis of high-risk varices in patients with NCPH [59]. Serum vitamin B12 levels have been found to be lower in IPH patients compared to those with cirrhosis, but this diagnostic potential requires further validation.

New biomarkers are gradually taking a role in clinical practice including anti-cell antibodies endothelial lesions (AECAs), as they have been shown to be frequently present in patients with NCPH. However, 16% of patients with cirrhosis also have a positivity to these autoantibodies; therefore, they cannot be used as a single parameter for the diagnosis even if it can guide toward a suspicion of NCPH [60].

Anti-endothelial cell antibodies have been suggested as a diagnostic marker, but they require validation [60,61].

However, several limitations to the definition of NCPH have been recognized as portal vein thrombosis and the coexistence of other conditions can complicate the diagnosis. Additionally, there is no specific diagnostic test available, and the current criteria rely on clinical signs of portal hypertension, which may not be present in early stages of the disease identified through liver biopsy. Finally, as also reported by the VALDIG group, the diagnosis of NCPH requires the histological exclusion of cirrhosis; furthermore, it requires either a specific sign of portal hypertension (gastric-esophageal varices, variceal bleeding, systemic collaterals detectable on imaging), or a specific sign of histological NCPH, or a non-specific sign of portal hypertension (ascites, platelets < 150,000/mm^3^, largest spleen diameter ≥ 13 cm) associated with non-specific histological signs of NCPH [21].

The diagnosis of NCPH excludes conditions affecting the hepatic veins (Budd–Chiari syndrome) and liver diseases causing microvascular damage (sarcoidosis, congenital hepatic fibrosis, sinusoidal obstruction syndrome).

Furthermore, myeloproliferative diseases should also be included in the differential diagnosis with hypertension.

Myeloproliferative diseases (MPDs), which involve the abnormal growth of hematologic cell lines leading to extra-medullary hematopoiesis, primarily in the spleen and sometimes in the liver, can also result in portal hypertension. This is believed to arise mainly due to left-sided portal hypertension. While true incidences of MPD leading to portal hypertension are likely underestimated, our findings resonate with other Asian studies. Interestingly, in a study conducted by Sharma M et al., portal vein thrombosis was not observed, but MPD patients displayed significant splenomegaly, causing increased blood flow through the splenic vein, which can trigger left-sided portal hypertension. Additionally, myeloid metaplasia in the liver may also contribute to right-sided portal hypertension [62].

## 8. Management and Treatment

The management of patients with NCPH aims to prevent and treat variceal bleeding. Standard approaches used in portal hypertension due to cirrhosis, such as nonselective beta blockers and endoscopic variceal ligation, are generally employed for the primary and secondary prevention of variceal bleeding [63].

Additional management measures include the discontinuation of NCPH-associated drugs and treatment of related medical conditions. Discontinuing medications like azathioprine in organ transplant recipients has shown improvements in biochemical and histological parameters [64].

Transjugular intrahepatic porto-systemic shunt (TIPS) is a valid option for patients with uncontrolled variceal bleeding despite medical and endoscopic treatment. Studies have reported favorable outcomes with a 2-year survival rate of 80%. Patients with extrahepatic comorbidities and elevated creatinine may face higher mortality risks [53].

Patients with NCPH have fewer complications related to TIPS compared to patients with cirrhosis. Lv et al. compared these two groups following TIPS insertion for the management of variceal bleeding, showing how patients with NCPH had significantly lower rates of hepatic encephalopathy, hepatic impairment, and long-term mortality [65].

We have seen how IPH is often associated with splenomegaly, thrombocytopenia and the presence of esophageal or gastric varices. Partial splenic artery embolization (PSE) is sometimes offered as a relatively safe alternative to splenectomy for hypersplenism. The majority of previous reports on patients with IPH undergoing PSE are case series. The largest one, reported by Romano et al. [66], had only six patients. In some patients, there was an improvement in pancytopenia by decreasing splenic volume and furthermore a reduction in variceal bleeding. Subsequent PSE can be performed with good results.

We report the case of a 34-year-old male patient suffering from IPH with splenomegaly and thrombocytopenia (70,000/mm^3^). He underwent selective embolization of the lower pole of the spleen without post-procedure complications. At one-month follow-up, there was an increase in number of platelets (/mm^3^) as well as a reduction in spleen volume (Figure 3).

In comparison with PSE, splenectomy is associated with more frequent major and minor complications. However, the effect on improvement in cell lines is only transient, and post-embolization syndrome is almost universal [67].

Karagul et al. in their review reported that eleven patients with NCPH who did not require liver transplantation were successfully operated on with a porto-systemic shunt procedure [68]. After failure of drug and endoscopic therapy, if liver function is well preserved, the shunt is the best therapeutic option for the prevention of variceal bleeding [69]. If ascites is unresponsive to medical treatment, a side-to-side portacaval shunt may be necessary. These shunts decompress both the splanchnic viscera and the hepatic sinusoids, effectively reducing ascites. This type of shunt has a high long-term patency rate, making it an effective option for managing refractory ascites [70]. Distal splenorenal shunts are projected to continue the perfusion of portal flow through the portal vein and are aimed at decreasing the risk of porto-systemic encephalopathy. Unfortunately, these patients tend to present postoperative ascites procedures [71]. As a bridge to transplant, a side-to-side PSS (either mesocaval or portocaval) can be performed if the retrohepatic vena cava is patent (Figure 4).

Liver transplantation is considered in patients with NCPH who meet the criteria for end-stage liver disease. The primary indication for liver transplantation is medically refractory severe portal hypertension. However, data on liver transplant outcomes in NCPH are limited, and the risk of NCPH relapse after transplantation is not well defined. Despite this, small case series suggest favorable survival rates. It is worth nothing that some cases of recurrent NCPH have been reported following liver transplantation [72] (Figure 4).

## 9. Conclusions

NCPH is probably a little-known and underestimated condition in the Western world. Physicians should learn to look for this condition in a variety of clinical settings, including cryptogenic cirrhosis, diseases known to be associated with NCPH, autoimmune diseases, drug administration and even alterations in coagulation function tests. Once NCPH is clinically suspected, liver histology has become mandatory for proper diagnosis. However, pathologists should be familiar with the histological features of NCPH, and a number of issues remain to be clarified, including NCPH etiology and pathogenesis, as well as its natural history, prognosis and management.

NCPH is an uncommon disorder that is often difficult to diagnose due to its varied presentation, clinical features and unusual imaging findings. Patients with NCPH are often clinically and radiologically misdiagnosed as liver cirrhosis, so a liver biopsy is indispensable to discriminate cirrhosis from NCPH.

Even if NCPH has been considered a disorder with a relatively benign disease course, systemic fatal complications can occur such as liver failure, hepatic encephalopathy, and hepatopulmonary syndrome. Therefore, we believe that this review can help heighten a sense of awareness of NCPH allowing earlier diagnosis, recognition of its complications and the correct management of it. Physicians should learn to look for this condition in a variety of clinical settings and to take into account the consequence of its therapeutic management, regarding the different underlying etiologies.

However, further therapeutic options suggested by different physician expertise, in order to obtain higher efficacy, better tolerance and fewer side effects are needed. We conducted the narrative review focusing on the most recent advances; however, the number of cases in the studies considered is still modest. Further research regarding the multidisciplinary management of NCPH is required.

## Figures and Tables

**Figure 1 diagnostics-13-03263-f001:**
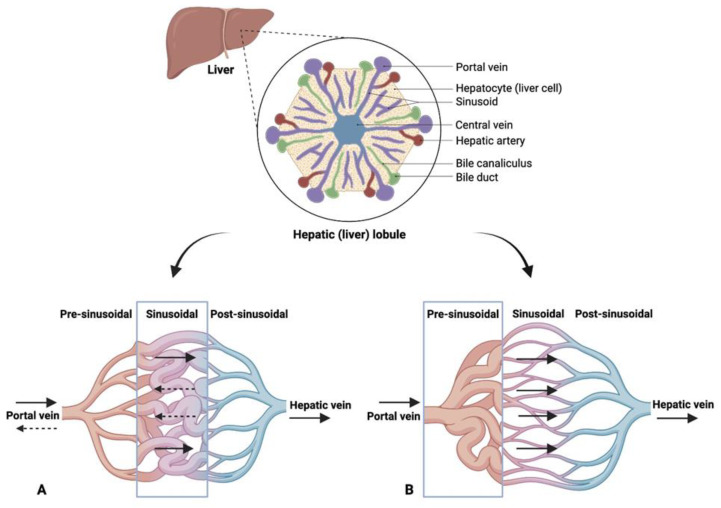
Differences between liver cirrhosis (**A**) and NCPH (**B**). In the first (**A**), an increase in resistance at the sinusoidal level and a hepatofugal flow is evident. In the second (**B**), the resistances increase at the presinusoidal level (created with BioRender.com, accessed on 11 October 2023).

**Figure 2 diagnostics-13-03263-f002:**
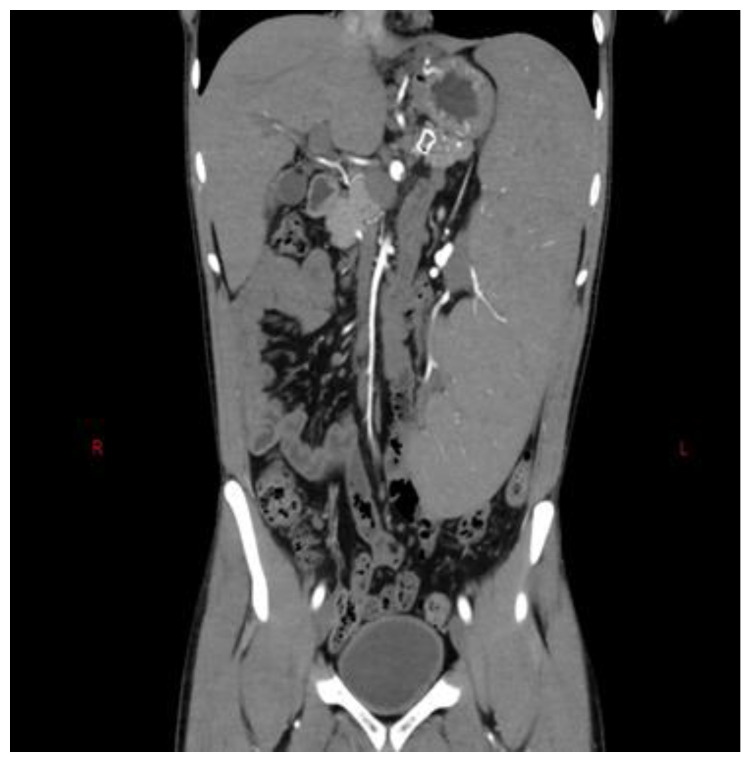
Patient with NCPH and associated splenomegaly.

**Figure 3 diagnostics-13-03263-f003:**
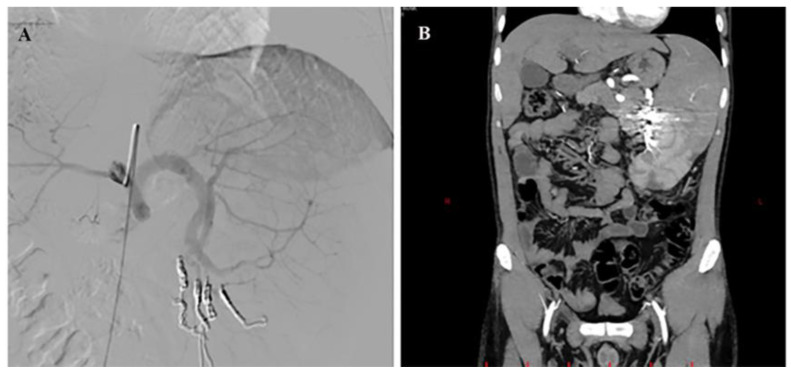
Super-selective catheterization of the inferior branches suppling the lower pole of the spleen (**A**) and post-embolization computer tomography (**B**) in patient with NCPH. The branch leading to the inferior pole of the splenic artery was selectively catheterized with a microcatheter and then embolized with coils of appropriate diameter. The branches leading to the upper lobe have been preserved.

**Figure 4 diagnostics-13-03263-f004:**
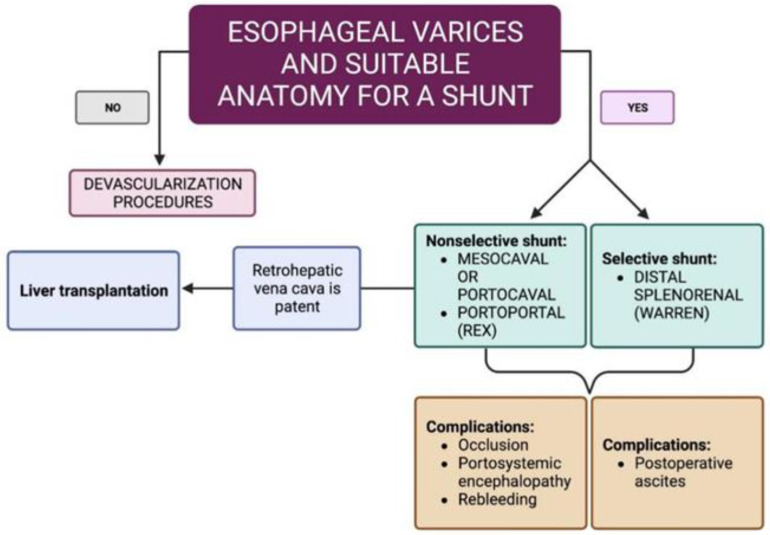
Surgery in patients with non-cirrhotic portal hypertension (created with BioRender.com, accessed on 16 May 2023).

**Table 1 diagnostics-13-03263-t001:** Hemodynamics of NCPH in comparison with liver cirrhosis.

	NCPH	Liver Cirrhosis
Portal venous blood flow	increased	normal
Intrahepatic vascular resistance at the sinusoidal level	decreased	increased
Intrahepatic vascular resistance at the presinusoidal level	increased	decreased
Hepatic arterial blood flow	decreased	increased
Arterioportal shunts	negligible	many
Hepatic vein-to-hepatic vein anastomoses	frequently	rare

## Data Availability

Not applicable.

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
