# Peer review of "A Narrative Review on Non-Cirrohotic Portal Hypertension: Not All Portal Hypertensions Mean Cirrhosis"

_diagnostics, 2023, doi:10.3390/diagnostics13203263_

Round 1

Reviewer 1 Report

Dear Author

I have the following comments

1. In international fraternity, PSVD is not preffered now. Rather, the term NCPH (Non-cirrhotic portal hypertension) is used. You may consider to us ethe same term in your manuscript

2. Epidemiology section: The data provided suggests 'prevalence' but tnot the incidence. 

3. P2L88:  According to classical teaching and scientific vocabulary, resistance in cirrhosis is at sinusoidal level but not the post-sinusoidal level. Post sinusoidal resistance is seen in Budd-Chiary syndrome or ther similar conditions.

4. Add a section on natural history of NCPH

5. Figure need to be more cleaer to understand. It is mot easy to understand. 

none

Author Response

Manuscript ID diagnostics-2611414

Response to Reviewer 1 Comments

  1. Summary

We thank the reviewer for their careful reading of the manuscript and their constructive remarks. Please find below a detailed point-by-point response to all comments (reviewers’ comments in black, our replies in blue).

  1. Point-by-point response to Comments and Suggestions for Authors.

Comments 1: In international fraternity, PSVD is not preferred now. Rather, the term NCPH (Non-cirrhotic portal hypertension) is used. You may consider to use the same term in your manuscript.

Response 1: We appreciate very much for the constructive suggestion. We have replaced the term “PSVD” with “NCPH”. However, to clarify why we chose to use the term “PSVD”, we suggest a recent review by De Gottardi A. et al., 2022 where authors explain how “The introduction of PSVD as a novel clinical entity will facilitate collaborative studies and investigations into the underlying molecular pathomechanisms encompassed by this term”.

Comments 2: Epidemiology section: The data provided suggests 'prevalence' but not the incidence.

Response 2: We thank the reviewer for pointing this out. We have reworded the statement to be more accurate (line 75-76).

Comments 3: P2L88: According to classical teaching and scientific vocabulary, resistance in cirrhosis is at sinusoidal level but not the post-sinusoidal level. Post sinusoidal resistance is seen in Budd-Chiary syndrome or their similar conditions.

Response 3: We thank the reviewer for highlighting this point. We have replaced the term “post-sinusoidal“ with “sinusoidal” (line 96). The same replace has been made in figure 1 legend.

Comments 4: Add a section on natural history of NCPH.

Response 4: A new section on natural history of NCPH was added (P6, line 257-296).

Comments 5: Figure need to be more clear to understand. It is not easy to understand.

Response 5: We apologize for any confusion. We modified the Figure 1 hoping it will be clearer and more comprehensive, according with text.

Reviewer 2 Report

This is a review of portal hypertension due to noncirrhosis and is well organized.

We believe that several modifications are desirable.

1. The description of histological features and diagnosis should be included. Please also include an educational figure if possible.

2. Splenomegaly and portal hypertension due to myeloproliferative disorders could be included in this review.

3. As per line 216, it would be better to mention more specific drug names and mechanisms of drug effects.

4. The splenic artery embolization in the treatment is not mentioned enough; please include a more specific description in the text, including an explanation of figure 4.

Author Response

Manuscript ID diagnostics-2611414

Response to Reviewer 2 Comments

  1. Summary

We thank the reviewer for their comments concerning our manuscript. Their comments are all valuable and very helpful for revising and improving our paper, as well as the important guiding significance to our researches. Please find below a detailed point-by-point response to all comments (reviewers’ comments in black, our replies in blue).

  1. Point-by-point response to Comments and Suggestions for Authors

Comments 1: The description of histological features and diagnosis should be included. Please also include an educational figure if possible.

Response 1: We thank the reviewer for his input, we believe that has been invaluable to make our review more balanced. We added a new section of “Histological features” as sub chapter of “Diagnosis” (P7, line 311-344).

Comments 2: Splenomegaly and portal hypertension due to myeloproliferative disorders could be included in this review.

Response 2: We appreciate very much for the constructive suggestion. We expanded this discussion with Myeloproliferative disorders” in the section “7.2 Clinical implication” (P7, line 388-399).

Comments 3: As per line 216, it would be better to mention more specific drug names and mechanisms of drug effects.

Response 3: We thank the reviewer for highlighting this point and we expanded this discussion (line 236-244).

Comments 4: The splenic artery embolization in the treatment is not mentioned enough; please include a more specific description in the text, including an explanation of figure 4.

Response 4: We appreciate very much for the constructive suggestion. A more specific description of the splenic artery embolization in the treatment was included in the text (line 421-436), and in the figure 3 legend.

Round 2

Reviewer 1 Report

Thanks for accepting my suggestions

Need minor modifications